**Data Availability Statement:** Data cannot be shared publicly because of the policy of Ethical

# Salivary and serum interleukin-17A and interleukin-18 levels in patients with type 2 diabetes mellitus with and without periodontitis

**Suteera Techatanawat[1][¤], Rudee Surarit[2], Kongthawat Chairatvit[2], Weerapan Khovidhunkit[3], Sittiruk Roytrakul[4], Supanee Thanakun[5,6], Hiroaki Kobayashi[7], Siribang-on Piboonniyom Khovidhunkit[8]\*, Yuichi Izumi[7,9]**

1 Ph.D. Program in Oral Biology, Faculty of Dentistry, Mahidol University, Ratchathewi, Bangkok, Thailand, 2 Department of Oral Biology, Faculty of Dentistry, Mahidol University, Ratchathewi, Bangkok, Thailand, 3 Department of Medicine, Faculty of Medicine, Chulalongkorn University, Pathumwan, Bangkok, Thailand, 4 National Center for Genetic Engineering and Biotechnology, National Science and Technology Development Agency, Khlong Luang, Pathum Thani, Thailand, 5 College of Dental Medicine, Rangsit University, Muang Pathum Thani, Pathum Thani, Thailand, 6 Oral Diagnosis and Oral Medicine Clinic, Dental Hospital, Faculty of Dentistry, Mahidol University, Ratchathewi, Bangkok, Thailand, 7 Department of Periodontology, Graduate School of Medical and Dental Sciences, Tokyo Medical and Dental University, Bunkyo-ku, Tokyo, Japan, 8 Department of Advanced General Dentistry, Faculty of Dentistry, Mahidol University, Ratchathewi, Bangkok, Thailand, 9 Oral Care Perio Center, Southern Tohoku General Hospital, Southern Tohoku Research Institute for Neuroscience, Koriyama, Fukushima, Japan

¤ Current address: Department of General Dentistry, Faculty of Dentistry, Srinakharinwirot University, Watthana, Bangkok, Thailand

\* siribangon.pib@mahidol.edu

## Abstract

### Objective

Interleukin (IL)-17A and IL-18 have been proposed to play important roles in periodontitis and type 2 diabetes mellitus (DM), but human data are conflicting. The present study aimed to investigate the roles of IL-17A and IL-18 in periodontitis and DM by measuring salivary and serum levels, respectively.

### Materials and methods

A total of 49 participants with type 2 DM and 25 control subjects without type 2 DM were recruited. A periodontal screening and recording (PSR) index (0, 1–2, 3, and 4) was used to classify whether these subjects had periodontitis. Salivary and serum IL-17A and IL-18 levels were measured by enzyme-linked immunosorbent assay. Multiple linear regression analyses were used to evaluate the associations between these cytokines and clinical parameters.

### Results

Salivary IL-17A levels were not significantly different between patients with DM and controls, however, the levels were significantly higher in controls with periodontitis than those without periodontitis ($p = 0.031$). Salivary IL-17A levels were significantly associated with the PSR

committee of the Faculty of Medicine, Chulalongkorn University. Data will be available from the Faculty of Medicine, Chulalongkorn University Institutional Data Access / Ethics Committee (contact via Miss Suwanna Muanpetch, aumaim44@hotmail.com) for researchers who meet the criteria for access to confidential data.

**Funding:** This study was supported primarily by a grant to S. Techatanawat and S.P. Khovidhunkit from the Thailand Research Fund through the Royal Golden Jubilee Ph.D. Program (Grant No. PHD/45/2556), and the funder had no role in study design, data collection and analysis, decision to publish, or preparation of the manuscript. This study was partially supported by Mahidol University under the National Research Universities Initiative to Rudee Surarit.

**Competing interests:** This study was supported primarily by a grant to S. Techatanawat and S.P. Khovidhunkit from the Thailand Research Fund through the Royal Golden Jubilee Ph.D. Program (Grant No. PHD/45/2556), and the funder had no role in the study design, data collection and analysis, decision to publish, or preparation of the manuscript. This does not alter our adherence to PLOS ONE policies on sharing data and materials, as all data and protocols underlying this study are shared, and the funding organization does not declare any bias towards particular study outcomes.

index ($\beta = 0.369$, $p = 0.011$). Multiple linear regression analyses revealed the association of salivary IL-18 levels and fasting plasma glucose ($\beta = 0.270$, $p = 0.022$) whereas serum IL-18 levels were associated with HbA$_{1C}$ ($\beta = 0.293$, $p = 0.017$). No correlation between salivary and serum levels of IL-17A and IL-18 was found.

## Conclusion

Salivary IL-17A was strongly associated with periodontitis, whereas salivary IL-18 was associated with FPG and serum IL-18 was associated with HbA$_{1C}$. These results suggest the role of these cytokines in periodontal inflammation and DM.

## Introduction

A close relationship between diabetes mellitus (DM) and periodontitis has long been recognized [1]. Patients with DM have an increased risk to develop periodontitis and those with untreated periodontitis seem to have a poorer glycemic control. Possible mechanistic links between DM and periodontitis have been proposed, including altered polymorphonuclear cell (PMN) function, increased adipokine production, and altered apoptosis, which could result in increased inflammatory cytokine production in both patients with periodontitis and DM [1]. Recent studies have identified inflammation as an important factor in the pathogenesis of DM [2, 3]. In clinical studies, increased levels of several pro-inflammatory cytokines including tumor necrosis factor-$\alpha$ (TNF-$\alpha$), interleukin (IL)-1, IL-6, and IL-18 were associated with various diabetic complications [4–6].

Local inflammation inevitably underlies the pathological basis of periodontitis. T-helper (Th) 17 cells, the recent subset of T-cells, were shown to be relevant to the pathogenesis of periodontal disease since increased Th17 cells in inflamed gingival tissue of periodontitis patients was demonstrated [7]. IL-17A is the most studied member of the Th17 cytokine family and its overproduction was related to autoimmune diseases and chronic inflammation, including periodontitis [8, 9]. IL-17A has been suggested to contribute to the pathogenesis of periodontitis in many ways. First, it can induce the receptor activator of nuclear factor κB (RANK)—RANK ligand (RANKL) signaling pathway which promotes osteoclastogenesis [10, 11]. Second, it acts as a regulatory cytokine that induces inflammatory responses by stimulating the release of other inflammatory cytokines including IL-6, IL-8, and IL-1β from macrophages, epithelial, and fibroblastic cells [11]. Third, it participates in regulating some matrix metalloproteinases (MMPs) production that could lead to periodontal tissue destruction [12]. Th17 cells have been proven as the main source of IL-17 in both healthy and diseased gingiva [13]. Other cells in periodontal tissues, including macrophages, mast cells, neutrophils, natural killer T (NKT) cells, gamma-delta (γδ) T-cells, and periodontal ligament cells can also produce this pro-inflammatory cytokine [14]. Therefore, periodontal inflammation could induce increased IL-17 levels. In addition, IL-18, a member of IL-1 family, is a potent inflammatory cytokine that regulates the inflammatory process by stimulating Th1 or Th2 responses. It can stimulate Th1 synergistically with IL-12 and results in the production of interferon-gamma (IFN-gamma) [15]. IL-18 is produced in various cell types, including endothelial cells, vascular smooth muscle cells, macrophages, dendritic cells, and adipocytes. It has also been suggested to be involved in periodontal inflammation since elevated IL-18 levels in gingival crevicular fluid (GCF) and saliva were found in patients with chronic periodontitis [16, 17].

Nevertheless, recent studies reported conflicting results regarding the roles of IL-17A and IL-18 in periodontal disease. Awang *et al.*[18] revealed significantly higher IL-17A levels in

GCF, saliva, and serum of subjects with periodontitis compared to those of subjects without periodontitis. The study of Esfahrood et al.[19] revealed no significant difference in both salivary and GCF levels of IL-18 between subjects with chronic periodontitis and those with healthy periodontium. In contrast, Ozcaka et al.[16] found significantly decreased salivary IL-17A and increased salivary IL-18 levels in subjects with chronic periodontitis compared to those without periodontitis. Therefore, it is still inconclusive whether periodontal inflammation results in increased IL-18 and IL-17A levels in oral environment.

In addition to the potential role of IL-17A and IL-18 in the local involvement of periodontitis, a systemic role of these cytokines and interactions with DM has also been recognized. In rodents, DM has been shown to enhance the mRNA expression and the protein levels of IL-17 in several tissues [20, 21]. In human, however, Roohi et al.[22] revealed similar levels of IL-17A in serum of both patients with type 1 DM and those with type 2 DM compared to healthy controls. Elevated serum IL-18 levels were reported in patients with metabolic syndrome and type 2 DM [15, 23]. Human data on serum and salivary IL-17A and IL-18 in patients with DM with and without periodontitis are, however, still sparse.

Therefore, this study was aimed to investigate the local and systemic role of IL-17A and IL-18 by measuring salivary and serum levels in subjects with type 2 DM and periodontitis. In addition, the correlation and association of these cytokines levels with clinical oral and systemic parameters were examined.

## Materials and methods

### Study population

Forty-nine subjects with type 2 DM were randomly recruited from the Outpatient Clinic of the Endocrinology and Metabolism Section, Department of Medicine, King Chulalongkorn Memorial Hospital, Bangkok, Thailand. Inclusion criteria comprised of those who were diagnosed with type 2 DM at least one year before and were on glycemic control medication and/or insulin. Exclusion criteria comprised of those who received periodontal treatment or antibiotic medication within the previous 6 months, had a history of salivary gland pathology or received radiotherapy at the head and neck area, had a number of teeth in an oral cavity less than 10, and had any clinical signs of intraoral inflammation or acute infection other than periodontal disease. Subjects who smoked, consumed alcohol beverages, were pregnant or lactating were also excluded. Twenty-five systemically healthy subjects who came to the hospital for routine medical check-up were recruited as control subjects. The same exclusion criteria were used for the control subjects. Each subject understood the protocol of this research study and signed a written inform consent. Prior to research commencement and subject enrollment, the protocol was approved by the Institutional Review Board, Faculty of Medicine, Chulalongkorn University (IRB No.100/57).

### Clinical and laboratory examinations

Demographic data and medical history were recorded using a review of the hospital chart and a questionnaire. All subjects underwent blood collection for the measurement of fasting plasma glucose (FPG) and HbA$_{1C}$. Since some reports indicated that kidney function could affect the IL-17A and IL-18 levels, we examined serum creatinine and the glomerular filtration rate (GFR) and used these for the adjustment in multiple linear regression analysis. The GFR which reflects overall kidney function was estimated from serum creatinine level using the re-expressed Modification of Diet in Renal Disease (MDRD) equation with Thai racial factor [24].

Intraoral examinations including periodontal screening and salivary flow rates were performed by one examiner to screen overall oral health status. All subjects were examined for their periodontal status using a periodontal screening and recording (PSR) index [25]. To investigate the PSR score, the mouth was divided into 6 sextants and all teeth were probed at 6 different sites (mesio-buccal, mid-buccal, disto-buccal, mesio-lingual, mid-lingual and disto-lingual) with the WHO periodontal probe containing a colored band (3.5–5.5 mm) and a 0.5 mm sphere on its active end. The PSR score was assessed using the following criteria: 0 if probing depth (PD) $< 3.5$ mm, no bleeding on probing (BOP), and no calculus; 1 if PD $< 3.5$ mm, presence of BOP, no calculus; 2 if PD $< 3.5$ mm, presence of BOP and calculus; 3 if PD is 3.5 to 5.5 mm; and 4 if PD $> 5.5$ mm. Only the highest score of each sextant was recorded as the sextant score. The maximum PSR score of all sextants was used to represent the overall periodontal status of each subject. In this study, subjects with a maximum PSR score of 3 or 4 in at least one sextant were considered to have periodontitis and those with a maximum PSR score of 0 to 2 were considered not to have periodontitis. According to a study by Primal and colleagues in 2014 [25], the predictive potential of the PSR score for the evaluation of periodontal disease was performed. The correlation of PSR score and AAP disease categories was examined and a significant correlation of PSR scores with periodontal disease ($R^2 = 0.43$, $p < 0.0001$) was observed. PSR scored a fairly accurate predictor of AAP disease category with area under receiver-operator curve = 0.73, $p < 0.0001$, hence, the PSR scores were used for the evaluation of periodontal status in this present study.

We first recruited subjects according to their glycemic conditions and they comprised the control group [C group (n = 25)] and the type 2 diabetic group [DM group (n = 49)]. These subjects were then allocated according to their periodontal status which encompassed the control subjects without periodontitis [C-NP (n = 17)]; the control subjects with periodontitis [C-P (n = 8)]; the type 2 DM subjects without periodontitis [DM-NP (n = 26)], and the type 2 DM subjects with periodontitis [DM-P (n = 23)]. Further categorization using only the maximum PSR score was performed in order to investigate only the effect of periodontal status on the cytokine levels. The flowchart demonstrating subject recruitment and categorizations is depicted in Fig 1.

## Sample collections and preparations

Serum and saliva were collected between 9:00 AM to 12:00 PM on the same day after an overnight fast. The unstimulated whole saliva was collected using the standard method described by Navazesh et al.[26]. Briefly, subjects were asked to spit the saliva approximately 5 mL into a sterile tube while placing that tube on ice. Protease inhibitor cocktail (Roche Diagnostics GmbH, Mannheim, Germany) was added immediately after saliva collection. Saliva was subsequently centrifuged at 10,000 g, 4°C for 10 mins to collect only the supernatant which was further aliquoted and stored at -80°C. Approximately 10 mL blood samples were collected by venipuncture at the area of antecubital fossa. Serum was isolated using serum-separating blood collection tubes (BD vacutainers® Plus Plastic Serum Tubes; BD Medical, Franklin Lakes, NJ, US). The collected whole blood samples were allowed to clot at room temperature for 30–45 mins and the serum was isolated by centrifugation at 1,000 g, 4°C for 15 mins. The serum samples were further aliquoted and stored at -80°C until assayed.

## IL-17A and IL-18 measurements in saliva and serum

IL-17A and IL-18 concentrations in the saliva and serum samples were determined using commercially available enzyme-linked immunosorbent assay (ELISA) kits for human IL-17A (Diaclone, Besancon Cedex, France) and human IL-18 (LifeSpan BioSciences, Seattle, USA). Both kits employed sandwich ELISA principle. The samples were thawed at room temperature and

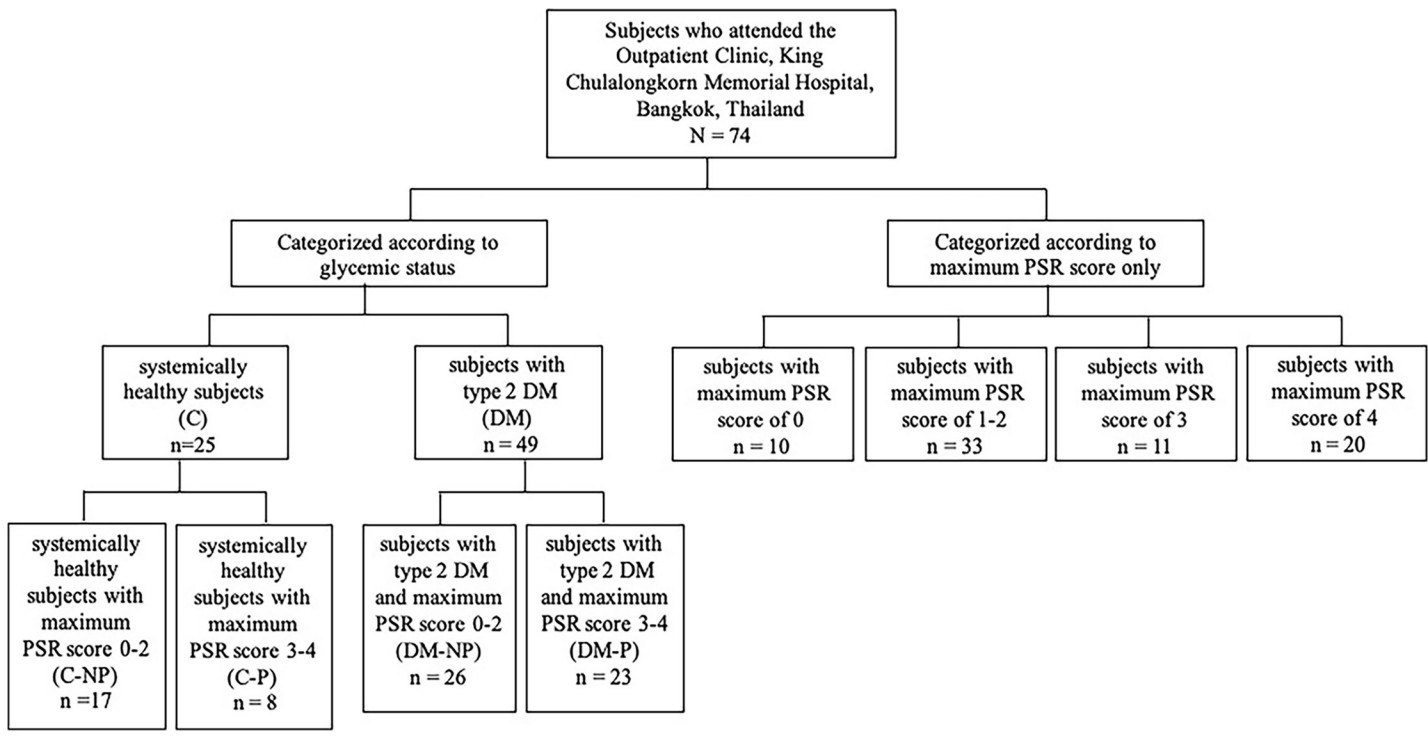

**Fig 1. Study flowchart for subject recruitment and categorizations.**

assayed according to the manufacturer's instructions. Different dilutions of sample were applied depending on the sensitivity of ELISA kits and different amounts of IL-17A and IL-18 in saliva and serum. 100 μL of serum samples at 2-fold dilution or undiluted saliva samples were used in IL-17A measurements. In IL-18 assays, 100 μL of serum or saliva samples at 50-fold dilution were used. Briefly, 100 μL of standard, sample or blank was added into each well, incubated for 90–120 mins at room temperature. After washing steps, 100 μL of biotinylated detection antibodies were added into each well and incubated for 1 hr at 37˚C. Then, after removing excess antibody by washing, 100 μL of HRP-streptavidin conjugate was added and incubated for 45 mins at 37˚C. TMB substrate solution was then added and incubated for 15–30 mins at 37˚C in the dark. Finally, 100 μL of stop solution was added into each well. The absorbance of samples and standards at 450 nm were measured immediately using VMax Kinetic ELISA Microplate Reader (Molecular Devices Corporation, Sunnyvale, CA, USA). A standard curve was organized in each experiment by SoftMax Pro Data Acquisition and Analysis Software (Molecular Devices Corporation) equipped in the microplate reader. The concentrations of IL-17A and IL-18 in each sample were determined based on standard curves and expressed in pg/mL. The lower detection limit for IL-17A and IL-18 was 2.3 pg/mL and 0.31 pg/mL, respectively.

## Statistical analyses

Sample size calculation for the DM and the control groups was performed using the power analysis program (G*power version 3.1, Dusseldorf, Germany). The total sample size of 74 (25 subjects who were systemically healthy and 49 subjects with type 2 DM), at significance level of 5% and effect size (f) of 0.5, could give the statistical power above 95% to detect the differences of cytokines levels between the study groups.

Statistical analyses were performed with SPSS 18.0 software (SPSS Inc., Illinois, USA). The normality of all continuous data was analyzed using the Shapiro-Wilk. The chi-squared test was applied to analyze the categorical variable. Comparisons of characteristics between controls and subjects with DM were performed using the independent T-test or Mann-Whitney $U$ test where appropriate. The Kruskal-Wallis test was used to compare characteristics of 4 groups according to periodontal status (control subjects without periodontitis, control subjects with periodontitis, type 2 DM subjects without periodontitis, and type 2 DM subjects with periodontitis) since the normality assumption was not satisfied. Data were presented as medians with the $1^{st}$ and $3^{rd}$ quartiles (if continuous) or frequency and percentage (if categorical). For cytokine levels analyses, non-parametric statistical analysis was utilized and data were presented as medians and interquartile ranges. Differences among 4 groups according to PSR index (PSR0, PSR1-2, PSR3, and PSR4) were assessed using the Kruskal-Wallis test and the Mann-Whitney $U$ test was further applied for post-hoc analyses. Partial correlation analyses adjusted for age and sex were applied to study the correlation between cytokine levels and clinical parameters. Stepwise multiple linear regression analyses were further performed to test the association of cytokine levels and clinical parameters. $P$ value of $< 0.05$ was considered statistically significant for all analyses.

## Results

### Characteristics of subjects

Forty-nine subjects with type 2 DM and 25 systemically healthy subjects were recruited. Although the number of subjects in the 2 groups was not similar, power of analysis indicated that the total sample size of 74 could give the power above 95% to detect the differences of cytokine levels between the 2 groups. We found that subjects in the DM group were significantly older, had significantly higher BMI, FPG, and $HbA_{1C}$ than those in the control group, whereas the salivary flow rate was significantly lower in the type 2 DM patients compared to the control subjects (Table 1). We further categorized these subjects into 4 groups according to periodontal status (control subjects without periodontitis, control subjects with periodontitis, type 2 DM subjects without periodontitis, and type 2 DM subjects with periodontitis) and found that there were significant differences in age, FPG, $HbA_{1C}$, and salivary flow rate among the 4 groups (Table 1).

### Levels of salivary IL-17A were strongly associated with periodontitis

We first examined salivary IL-17A levels in the control group compared to the DM group but no significantly different levels of this cytokine were found (Fig 2A). However, when the periodontal status was taken into account, significantly higher salivary IL-17A levels in the control subjects with periodontitis (C-P group) compared to those without periodontitis (C-NP group) ($p = 0.031$) was observed (Fig 2E). There was a trend of higher levels of salivary IL-17A in the type 2 DM subjects with periodontitis (DM-P group) than that of the type 2 DM subjects without periodontitis (DM-NP group) but no statistical significance was revealed ($p > 0.05$) (Fig 2E).

When we further classified all subjects into 4 groups according to their maximum PSR scores (PSR0, PSR1-2, PSR3, and PSR4), we found that salivary IL-17A levels were significantly increased in subjects with PSR4 than those with PSR0 ($p = 0.001$) and PSR1-2 ($p = 0.04$) (Fig 3). Moreover, when the correlations between salivary IL-17A levels and clinical parameters were determined, salivary IL-17A levels were found to be positively correlated with PSR index ($r = 0.329$, $p = 0.016$) after they were adjusted for age and sex (Table 2).

**Table 1. Characteristics of subjects categorized according to glycemic status (control and DM) and periodontal status (with or without periodontitis).**

| | Group according to glycemic status | | | Group according to periodontal status | | | | |
|---|---|---|---|---|---|---|---|---|
| | Control n = 25 | DM n = 49 | *p*-value* | C-NP n = 17 | C-P n = 8 | DM-NP n = 26 | DM-P n = 23 | *p*-value# |
| **Age** | 54.0 (47.0, 60.5) | 61.0 (55.5, 66.5) | **0.003**[a] | 54.0 (45.0, 59.0) | 55.5 (50.25, 65.50) | 61.0 (55.50, 65.25) | 63.0 (55.0, 69.0) | **0.021**[d] |
| **Female gender** | 21 (84%) | 31 (63.3%) | NS[b] | 14 (82.4%) | 7 (87.5%) | 19 (73.1%) | 12 (52.2%) | NS[b] |
| **FPG (mg/dL)** | 88.0 (82.5, 94.0) | 126.0 (102.5, 157.0) | **<0.001**[a] | 88.0 (82.5, 90.5) | 92.0 (83.0, 105.3) | 126.5 (109.3, 151.8) | 126.0 (99.0, 160.0) | **<0.001**[d] |
| **HbA$_{1C}$ (%)** | 5.5 (5.2, 5.7) | 7.3 (6.4, 7.9) | **<0.001**[c] | 5.6 (5.2, 5.7) | 5.5 (5.3, 5.8) | 7.3(6.7, 7.9) | 6.6 (6.0, 7.8) | **<0.001**[d] |
| **eGFR (mL/min/1.73m$^2$)** | 83.04 (73.6, 96.3) | 83.10 (72.1, 96.5) | NS[c] | 86.2 (70.8, 100.8) | 82.4 (77.7, 92.5) | 84.1 (74.1, 97.1) | 85.4 (68.6, 96.5) | NS[d] |
| **BMI (kg/m$^2$)** | 23.61 (21.6,26.6) | 25.97 (23.4, 28.0) | **0.017**[c] | 23.1 (20.6, 25.7) | 24.1 (22.4, 27.5) | 25.8 (24.2, 27.8) | 26.6 (23.1, 28.2) | NS[d] |
| **Salivary flow rate (mL/min)** | 0.30 (0.2, 0.6) | 0.19 (0.1, 0.3) | **0.008**[c] | 0.3 (0.2, 0.6) | 0.6 (0.3, 0.8) | 0.2 (0.1, 0.3) | 0.2 (0.1, 0.3) | **0.021**[d] |

Bolds denote statistical significance ($p < 0.05$) while NS represents no statistical significant difference. Values are presented as medians and interquartile ranges (1st, 3rd quartile). Categorical data are presented as counts with percent values within brackets. Control: systemically healthy subjects; DM: subjects with type 2 DM; C-NP: control w/o periodontitis; C-P: control with periodontitis; DM-NP: type 2 DM w/o periodontitis; DM-P: type 2 DM with periodontitis; BMI: body mass index; eGFR: estimated glomerular filtration rate; FPG: fasting plasma glucose; HbA$_{1C}$: glycated hemoglobin

* Comparison between the control and the DM groups

# Comparison among the C-NP, C-P, DM-NP, and DM-P groups

[a] Independent t-test

[b] Chi-squared test

[c] Mann-Whitney *U* test

[d] Kruskal-Wallis test

To identify factors that independently affected salivary IL-17A, stepwise multiple linear regression analyses were further performed using age, sex, HbA$_{1C}$, eGFR, and PSR index as independent variables and cytokine levels as dependent variables (Table 3). Moreover, we also analyzed the association of salivary IL-17A and clinical parameters using FPG, instead of HbA$_{1C}$, as an independent factor regarding glycemic status (Table 4). Salivary IL-17A levels were positively associated with PSR index ($\beta = 0.369$, $p = 0.011$) independent of age, sex, HbA$_{1C}$, and eGFR (Table 3). Subjects with higher PSR scores seemed to have higher salivary IL-17A levels. When we used FPG instead of HbA$_{1C}$ as an independent factor represented for glycemic status in stepwise multiple linear regression analysis, salivary IL-17A levels were still associated with PSR index ($\beta = 0.344$, $p = 0.015$) (Table 4). These confirmed that salivary IL-17A levels were strongly and persistently associated with PSR index and not type 2 diabetic condition.

## Levels of serum IL-17A and its association with clinical parameters

Salivary IL-17A levels may reflect the local role of this cytokine in the pathogenesis of periodontitis, whereas serum IL-17A levels may reflect the systemic role in a variety of conditions. We next measured serum IL-17A levels and found that serum IL-17A levels were not significantly different between the DM group and the control group or between those with and without periodontitis (Fig 2B and 2F). Partial correlation analysis exhibited no significant correlation between serum IL-17A levels and other clinical parameters when adjusted for age and sex (Table 2). When stepwise multiple linear regression analysis using HbA$_{1C}$ as an independent factor was applied, only marginal significant association of age and serum IL-17A was found ($\beta = -0.265$, $p = 0.048$) (Table 3). When FPG was used as an independent factor, no association of the levels of serum IL-17A and any clinical parameters were revealed (Table 4).

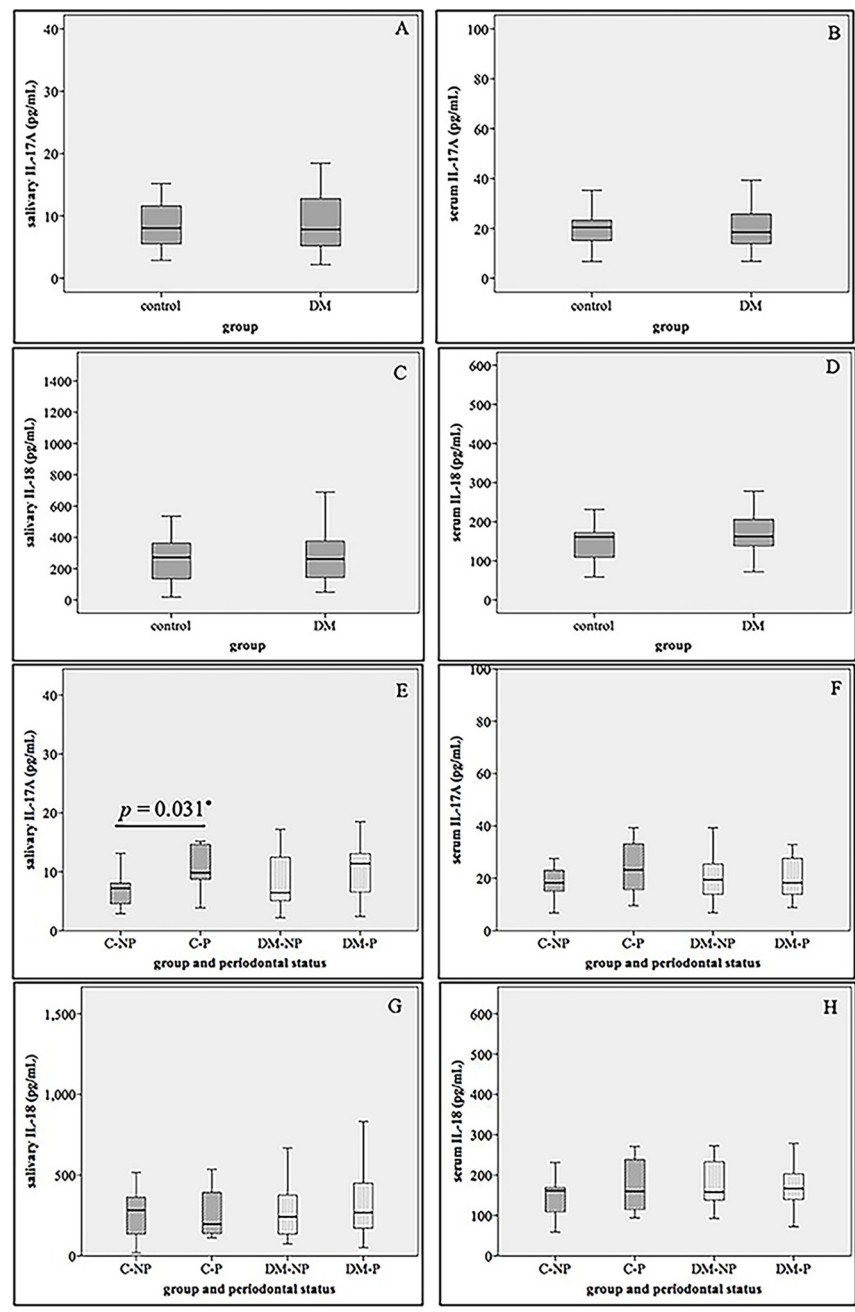

**Fig 2.** IL-17A and IL-18 levels of subjects categorized according to systemic (A-D) and periodontal health status (E-H). Salivary IL-17A (A and E); serum IL-17A (B and F); salivary IL-18 (C and G); serum IL-18 (D and H). Control: systemically healthy subjects; DM: subjects with type 2 DM; C-NP: control w/o periodontitis; C-P: control with periodontitis; DM-NP: type 2 DM w/o periodontitis; DM-P: type 2 DM with periodontitis. *Comparison between C-NP and C-P group was analyzed with the Mann-Whitney *U* test.

## Salivary IL-18 levels were associated with FPG

We initially examined salivary IL-18 levels in the controls and the type 2 diabetic patients and found that salivary IL-18 levels were not significantly different between the DM group and the control group (Fig 2C). Moreover, salivary IL-18 levels were not significantly different between those with and without periodontitis (Fig 2G). Interestingly, partial correlation analyses

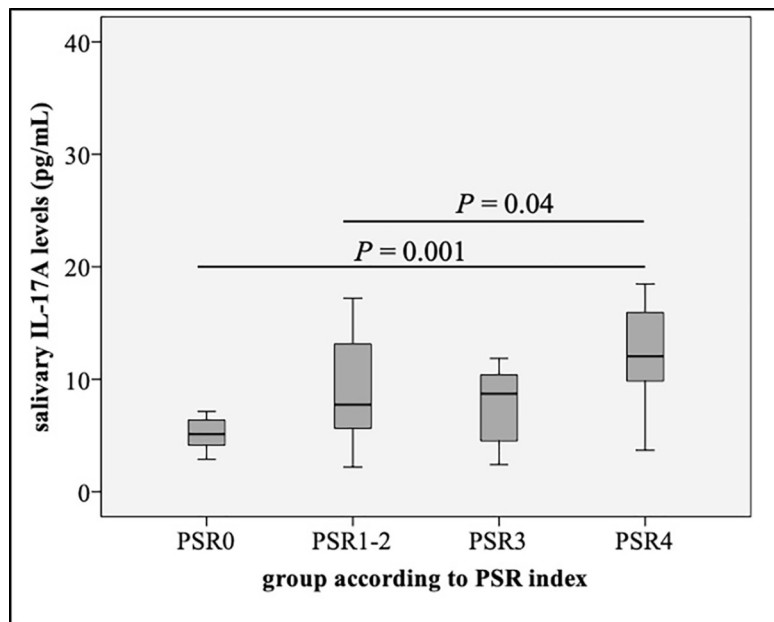

**Fig 3. Salivary IL-17A levels in all subjects categorized by PSR index.** Salivary IL-17A levels were significantly increased in subjects with PSR score 4.

revealed that salivary IL-18 levels were correlated with FPG ($r = 0.280$, $p = 0.017$) after they were adjusted for age and sex as shown in Table 2. In stepwise multiple linear regression analyses using $HbA_{1C}$ as an independent factor, only marginal significant association between salivary IL-18 and age was revealed ($\beta = 0.269$, $p = 0.049$) (Table 3). Notably, when we used FPG as an independent factor represented for glycemic status, we found that salivary IL-18 levels were positively associated with FPG ($\beta = 0.270$, $p = 0.022$) independent of age, sex, eGFR, and PSR index (Table 4). Subjects with higher FPG levels tended to have higher salivary IL-18 levels in our study.

### Serum IL-18 levels was associated with $HbA_{1C}$

Similar to salivary IL-18, serum IL-18 levels were not significantly different between the DM group and the control group or between those with and without periodontitis (Fig 2D and

**Table 2. Partial correlations adjusted for age and sex between cytokine levels and oral and clinical parameters.**

| Oral and clinical parameters | Salivary IL-17A (pg/mL) | Serum IL-17A (pg/mL) | Salivary IL-18(pg/mL) | Serum IL-18 (pg/mL) |
|---|---|---|---|---|
| **PSR index** | **r = 0.329** | r = 0.145 | r = 0.071 | r = -0.008 |
| | **p = 0.016** | p = 0.227 | p = 0.554 | p = 0.947 |
| **FPG (mg/dL)** | r = -0.114 | r = 0.108 | **r = 0.280** | r = 0.096 |
| | p = 0.418 | p = 0.369 | **p = 0.017** | p = 0.420 |
| **$HbA_{1C}$ (%)** | r = -0.036 | r = 0.040 | r = -0.043 | **r = 0.331** |
| | p = 0.800 | p = 0.743 | p = 0.724 | **p = 0.005** |
| **eGFR (mL/min/1.73 m$^2$)** | r = -0.182 | r = 0.064 | r = 0.194 | r = 0.013 |
| | p = 0.193 | p = 0.597 | p = 0.103 | p = 0.913 |

Bolds denote statistical significance. eGFR: estimated glomerular filtration rate; FPG: fasting plasma glucose; $HbA_{1C}$: glycated hemoglobin; PSR index: periodontal screening and recording index

**Table 3. Stepwise multiple linear regression analyses adjusted for age, sex, HbA$_{1C}$, eGFR, and PSR index were performed to investigate the association of clinical parameters with levels of IL-17A and IL-18 as continuous measures.**

| Variables | Cytokine levels (pg/mL) | | | | | | | |
|---|---|---|---|---|---|---|---|---|
| | Salivary IL-17A | | Serum IL-17A | | Salivary IL-18 | | Serum IL-18 | |
| | β | p | β | p | β | p | β | P |
| Age (years) | -0.090 | 0.552 | **-0.265** | **0.048** | **0.269** | **0.049** | -0.086 | 0.500 |
| Sex (female) | -0.004 | 0.978 | -0.018 | 0.882 | 0.009 | 0.940 | **-0.271** | **0.024** |
| HbA$_{1C}$ (%) | 0.078 | 0.588 | 0.061 | 0.624 | -0.032 | 0.798 | **0.293** | **0.017** |
| eGFR (mL/min/1.73m$^2$) | -0.245 | 0.107 | 0.075 | 0.563 | 0.201 | 0.125 | -0.034 | 0.780 |
| PSR index | **0.369** | **0.011** | 0.136 | 0.258 | 0.054 | 0.653 | 0.022 | 0.850 |

β values were derived from stepwise multiple linear regression analyses adjusted for age, sex, HbA$_{1C,}$ eGFR, and PSR index.

Bolds denote statistical significance ($p < 0.05$). eGFR: estimated glomerular filtration rate; HbA$_{1C}$: glycated hemoglobin; PSR index: periodontal screening and recording index.

2H). After partial correlation analysis was utilized, serum IL-18 levels were correlated with HbA$_{1C}$ ($r = 0.331$, $p = 0.005$) after they were adjusted for age and sex as demonstrated in Table 2.

Importantly, serum IL-18 levels were also associated with HbA$_{1C}$ ($β = 0.293$, $p = 0.017$) independent of age, sex, eGFR, and PSR index (Table 3). Subjects with high HbA$_{1C}$ levels seemed to have high serum IL-18 levels. Lower serum IL-18 levels were associated with the female gender ($β = -0.271$, $p = 0.024$) independent of age, HbA$_{1C}$, eGFR, and PSR index (Table 3). But this association was not found in the analysis using FPG as an independent factor (Table 4).

## Correlation between cytokine levels in saliva and serum

Furthermore, we investigated the correlation between cytokine levels in saliva and serum using partial correlation analyses. No significant correlation between salivary and serum levels of both IL-17A and IL-18 was revealed in this study (Fig 4).

In conclusion, we found that salivary IL-17A levels were positively associated with PSR index when age, sex, glycemic status, and kidney function were controlled. On the contrary, IL-18 levels reflected the glycemic status but not the periodontal condition. Salivary IL-18 levels were associated with FPG whereas serum IL-18 levels were associated with HbA$_{1C}$ when age, sex, kidney function, and periodontal status were controlled.

**Table 4. Stepwise multiple linear regression analyses adjusted for age, sex, FPG, eGFR, and PSR index were performed to investigate the association of clinical parameters with levels of IL-17A and IL-18 as continuous measures.**

| Variables | Cytokine levels (pg/mL) | | | | | | | |
|---|---|---|---|---|---|---|---|---|
| | Salivary IL-17A | | Serum IL-17A | | Salivary IL-18 | | Serum IL-18 | |
| | β | p | β | p | β | p | β | P |
| Age (years) | -0.057 | 0.695 | -0.239 | 0.064 | 0.227 | 0.071 | -0.089 | 0.494 |
| Sex (female) | -0.022 | 0.878 | -0.024 | 0.849 | 0.062 | 0.598 | -0.204 | 0.101 |
| FPG (mg/dL) | -0.035 | 0.799 | 0.106 | 0.375 | **0.270** | **0.022** | 0.095 | 0.430 |
| eGFR (mL/min/1.73m$^2$) | -0.240 | 0.113 | 0.054 | 0.673 | 0.188 | 0.137 | 0.009 | 0.948 |
| PSR index | **0.344** | **0.015** | 0.130 | 0.274 | 0.054 | 0.637 | -0.010 | 0.936 |

β values were derived from stepwise multiple linear regression analyses adjusted for age, sex, FPG, eGFR, and PSR index.

Bolds denote statistical significance ($p < 0.05$). eGFR: estimated glomerular filtration rate; FPG: fasting plasma glucose; PSR index: periodontal screening and recording index

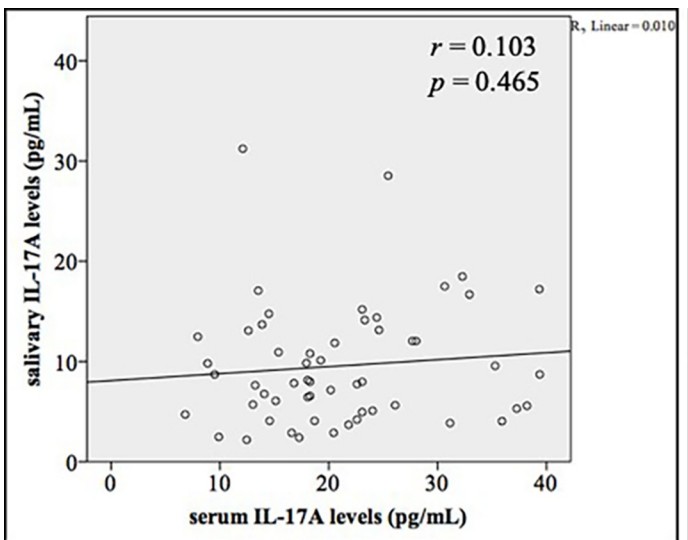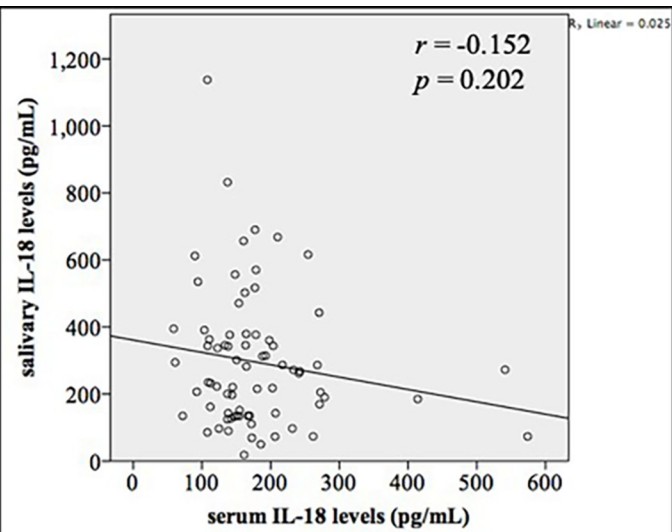

**Fig 4. Partial correlation between salivary and serum levels of IL-17A and IL-18 after adjustment for age and sex.** No significant correlation was revealed between salivary and serum levels of IL-17A and IL-18.

## Discussion

Both IL-17A and IL-18 have been proposed to be biomarkers for periodontitis [14, 16] and were linked to the pathogenesis of type 2 DM as well [9, 22, 27]. Both diseases share a common characteristic of chronic inflammation resulting from host immune responses [1]. The present study aimed to investigate the salivary and serum levels of inflammatory cytokines IL-17A and IL-18 in subjects with different glycemic and periodontal status in order to clarify the interrelationship among these cytokines, type 2 DM, and periodontitis.

Our study revealed that salivary IL-17A levels were firmly associated with the severity of periodontitis without the influence of glycemic status. Both periodontal disease and type 2 DM were recognized as chronic inflammatory diseases and a two-way relationship between them have been suggested [1, 2, 28]. DM could result in increased accumulation of the advanced glycation end products (AGEs) in body tissues, including gingiva [29]. This was proposed to contribute to the exaggerated immune responses and impaired tissue integrity which make diabetic patients more susceptible to periodontitis [30]. At first, we expected to see synergistic effects of these two diseases on the levels of this cytokine, however, we found no such effects on patients with type 2 DM who also had periodontal disease (DM-P group). The increasing trend of salivary IL-17A levels in subjects with periodontitis compared to those without periodontitis in both the control and the type 2 DM groups are demonstrated in our study (Fig 2E). This result was confirmed after all subjects were separated according to PSR index scores and significantly higher salivary IL-17A levels were detected in subjects with more severe periodontal disease compared to those with less severity (Fig 3). Importantly, after controlling the glycemic status, using $HbA_{1C}$ and FPG in stepwise multiple linear regression analyses, salivary IL-17A levels were still positively associated with PSR index score (Tables 3 and 4).

The reasons why salivary IL-17A levels did not relate with glycemic status may be explained as follows. Although glycemic status has been accepted to have an impact on periodontal tissues inflammation, it is not necessary to be reflected as an increased salivary IL-17A level. Our findings are correlated with the study of Gursoy *et al.*[31] which investigated salivary IL-17 concentrations in diabetic subjects with well-controlled ($HbA_{1C} < 7$) and poorly-controlled

(HbA$_{1C}$ ≥ 7) glycemic status and found no difference of salivary IL-17 levels between the 2 groups. They reported the association of periodontal probing depth and salivary IL-17 levels which was independent from glycemic status. Although subjects in the poorly-controlled glycemic status exhibited more severe periodontal conditions, they proposed that there might be some other underlying mechanisms rather than IL-17 to promote the periodontal destruction in poorly-controlled diabetic patients. In addition, subjects in the type 2 DM group in our study were on glycemic control medication and the HbA$_{1C}$ levels were considered as well-controlled glycemic status (HbA$_{1C}$ around 7% or below)(Table 1)[32]. Therefore, when the blood sugar was controlled, this might result in minimal changes of cytokines secreted in these patients and IL-17A might not play a pivotal role in inducing inflammation in type 2 DM unlike in periodontal inflammation. Another possible explanation is that the local inflammation due to periodontitis might affect more on the local cytokine levels rather than the systemic condition like type 2 DM. Taken together, these may obscure the influence of glycemic status on IL-17A levels in this study.

This study also demonstrated the benefit of saliva specimen in periodontal diagnostics since we found that salivary IL-17A levels could reflect the severity of periodontitis. Although, GCF has long been the specimen type for periodontal research [33], its collection requires specially trained professionals. In addition, GCF is originated from the gingival plexus of blood vessels in gingiva, transported through the epithelial lining along the dentogingival space and secreted into gingival sulcus either as a physiological fluid or inflammatory exudate depending on the condition of periodontal tissues [33, 34]. Thus, the composition of GCF can be varied due to periodontal inflammation. Subsequently, this fluid will be eluted into the whole saliva. Therefore, saliva specimens could harbor various molecular biological markers from GCF. This could explain why salivary IL-17A is upregulated in subjects with periodontitis.

The relationship between serum IL-17A and type 2 DM or periodontitis was not found in this study. This is consistent with the study of Ozcaka *et al*. which demonstrated that plasma IL-17 concentrations were similar between the chronic periodontitis group and the healthy control group [16]. However, one study showed higher serum IL-17A levels in patients with chronic periodontitis [18]. Therefore, there is still a controversy regarding the relationship between serum IL-17A levels and periodontal inflammation.

IL-18 levels were associated with glycemic status independently from periodontal condition. Although the role of IL-18 in periodontal disease has been addressed in recent studies [16, 35, 36], our study showed no significant difference in salivary IL-18 levels among subjects with or without type 2 DM and with or without periodontitis. This result is similar to the study of Correa *et al*.[37] which compared IL-18 levels in GCF of patients with type 2 DM to those of systemically healthy subjects, both of which were diagnosed with chronic periodontitis. They demonstrated no significant difference in GCF IL-18 levels between the two groups both at baseline and 90 days after non-surgical periodontal treatment. Moreover, they reported no significant difference of GCF IL-18 levels between pre- and post—periodontal treatment in healthy group with periodontitis. Differently, Ozcaka *et al*.[16] studied healthy subjects without any systemic diseases including type 2 DM and revealed significantly higher salivary IL-18 levels in subjects with chronic periodontitis compared to periodontal healthy subjects. Furthermore, no increased levels of serum IL-18 in the chronic periodontitis group was revealed. In another study, saliva and plasma samples were collected from 40 patients with chronic periodontitis and 20 healthy subjects. IL-18 measurements were performed using commercially available ELISA kits. Patients with chronic periodontitis exhibited 46% increase in plasma IL-18 levels compared to control subjects. Notably, salivary IL-18 levels rose up to > 5-fold in patients with chronic periodontitis compared with healthy individuals [38].

These contradictory results among these studies may be due to subject selection and also different methods used for the detection of IL-18 in saliva.

Interestingly, we found the association between salivary IL-18 levels and FPG independently from periodontal condition. This finding may be explained by the effect of hyperglycemic spikes on increasing inflammatory cytokines described by Esposito *et al.*[39]. In their study, acute hyperglycemic condition due to exogenous glucose injection could increase plasma and possibly lead to increased salivary IL-6, IL-18 and TNF-α levels in both healthy and impaired glucose tolerance (IGT) subjects and these increasing levels of cytokines were more pronounced in IGT subjects. Thus, higher salivary IL-18 level may result from the high plasma glucose level at that time point.

Serum IL-18 levels were associated with $HbA_{1C}$ but not periodontitis. The elevated serum IL-18 levels in type 2 DM and metabolic syndrome have been reported and have been suggested to contribute to microangiopathy in type 2 DM in recent studies [23, 27, 40]. In the present study, no significant difference of the serum IL-18 levels between healthy subjects and type 2 diabetic subjects was observed. However, our study could demonstrate the association between serum IL-18 and $HbA_{1C}$ independent from age, sex, eGFR, and PSR index. Other studies also reported the correlation between serum IL-18 and $HbA_{1C}$ [6, 41]. These results confirm the relationship between serum IL-18 and glycemic status. On the contrary, serum IL-18 level seemed to be independent from periodontal status. Our study showed no significant difference between serum IL-18 levels in periodontal healthy and periodontitis subjects and this correlates with the study of Ozcaka *et al.*[16] which demonstrated similar serum IL-17A and IL-18 levels between chronic periodontitis and periodontally healthy subjects.

Towards the interrelationship between salivary and serum IL-17A an IL-18 levels, we found no correlation between them. This result is in line with the study of Ozcaka *et al.*[16] which investigated plasma and salivary IL-17A and IL-18 levels in subjects with healthy periodontal status and chronic periodontitis and found no correlation. The regulation of cytokines in saliva and serum seemed to be different. Williamson *et al.* demonstrated that only 3 out of 27 cytokines were found to have the correlations between salivary and plasma levels and IL-17 levels in plasma and saliva were also reported to be uncorrelated [42]. The local immune response in oral environment might influence salivary cytokine components rather than the systemic immune response. The regulations of IL-17A and IL-18 in oral environment remained to be fully characterized to understand the role of IL-17A and IL-18 in oral biology.

We acknowledged that our study still has some limitations. First, the recruited subjects with type 2 DM were already on glycemic control medication and they were considered to exhibit well-controlled glycemic status [32]. Therefore, the effect of glycemic status on cytokine levels might not be obviously revealed and future study with the recruitment of patients with less controlled-glycemic status should be encouraged. Second, we used PSR score to screen for periodontal status of subjects. This screening index might not be as accurate compared to other periodontal parameters, however, some reports revealed convenience, reliability, reproducibility, and practicality in using this method to evaluate periodontal disease [43, 44]. Thus, we chose this practical index for the representation of periodontal status and still found that the levels of IL-17A were strongly associated with periodontal disease severity.

## Conclusion

Our study is the first to report the significant association of salivary IL-17A with PSR index without the influence of glycemic status. The elevated salivary IL-17A levels reflected more periodontal inflammation. On the contrary, salivary and serum IL-18 levels did not reflect periodontal inflammation, but instead correlated to the glycemic status.

## Acknowledgments

We would like to thank the staffs from the Oral Biology Department, Faculty of Dentistry, Mahidol University for their technical supports. S.P. Khovidhunkit is an Anandamahidol scholar.

## Author Contributions

**Conceptualization:** Supanee Thanakun, Siribang-on Piboonniyom Khovidhunkit.

**Data curation:** Suteera Techatanawat.

**Formal analysis:** Suteera Techatanawat, Supanee Thanakun.

**Investigation:** Suteera Techatanawat.

**Methodology:** Suteera Techatanawat, Siribang-on Piboonniyom Khovidhunkit.

**Project administration:** Siribang-on Piboonniyom Khovidhunkit.

**Resources:** Weerapan Khovidhunkit, Hiroaki Kobayashi, Yuichi Izumi.

**Supervision:** Rudee Surarit, Kongthawat Chairatvit, Weerapan Khovidhunkit, Sittiruk Roytrakul, Supanee Thanakun, Hiroaki Kobayashi, Siribang-on Piboonniyom Khovidhunkit, Yuichi Izumi.

**Validation:** Suteera Techatanawat, Supanee Thanakun, Siribang-on Piboonniyom Khovidhunkit.

**Visualization:** Suteera Techatanawat, Siribang-on Piboonniyom Khovidhunkit.

**Writing – original draft:** Suteera Techatanawat, Weerapan Khovidhunkit, Supanee Thanakun, Siribang-on Piboonniyom Khovidhunkit.

**Writing – review & editing:** Suteera Techatanawat, Rudee Surarit, Weerapan Khovidhunkit, Supanee Thanakun, Siribang-on Piboonniyom Khovidhunkit.

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
