## [Decision Letter · Decision Letter 0]

4 Sep 2019

PONE-D-19-15328

Salivary and serum interleukin-17A and interleukin-18 levels in patients with type 2 diabetes mellitus with and without periodontitis.

PLOS ONE

Dear  Dr. Khovidhunkit

Thank you for submitting your manuscript to PLOS ONE. After careful consideration, we feel that it has merit but does not fully meet PLOS ONE’s publication criteria as it currently stands. Therefore, we invite you to submit a revised version of the manuscript that addresses the points raised during the review process.

As suggested by the reviewers, there are some methodological points that should be addressed before the manuscript is acceptable for publication. SPecifically pay attention to the limitation of the study for the low number of patients studied, and the need for an appropriate calculation of sample size.

We would appreciate receiving your revised manuscript by November 30th. To enhance the reproducibility of your results, we recommend that if applicable you deposit your laboratory protocols in protocols.io, where a protocol can be assigned its own identifier (DOI) such that it can be cited independently in the future. For instructions see: http://journals.plos.org/plosone/s/submission-guidelines#loc-laboratory-protocols

We look forward to receiving your revised manuscript.

Kind regards,

Víctor Sánchez-Margalet

Academic Editor

PLOS ONE

Journal Requirements:

Additional Editor Comments (if provided):

Reviewers' comments:

Reviewer's Responses to Questions

**Comments to the Author**

1. Is the manuscript technically sound, and do the data support the conclusions?

Reviewer #1: No

Reviewer #2: Partly

2. Has the statistical analysis been performed appropriately and rigorously? 

Reviewer #1: No

Reviewer #2: Yes

3. Have the authors made all data underlying the findings in their manuscript fully available?

Reviewer #1: Yes

Reviewer #2: Yes

4. Is the manuscript presented in an intelligible fashion and written in standard English?

Reviewer #1: Yes

Reviewer #2: Yes

5. Review Comments to the Author

Reviewer #1: This paper looks at Il 17 and 18in DM and periodontitis.

I have a few comments listed below.

The introduction is long and could be reduced.

To be diagnosed with periodontitis how many sextants had to score 3 or 4? There was no radiographic confirmation of bone loss?

The healthy control group with periodontitis has very low numbers.

How and where from were blood samples collected?

Although a sample size calculation was undertaken it is just for two groups, not the four you have. Did you also age/sex match the groups?

I feel you need to repeat the sample size calculation and recruit more healthy periodontitis patients.

You need to add exactly what the laboratory methods were.

How many samples did not have detectable Il17 or 18?

The results introduce new methods with the classification by PSR. This is not what the study was designed for and may not be valid statistically.

The legends for Tables 3 and 4 need to better explain what they show.

The results are long and perhaps overanalyse the data

Given the issues with sample size a lot of the statements in the discussions are difficult to support.

Reviewer #2: Techatanawat et al examine interleukin-17A and interleukin-18 in paired saliva and blood samples from subjects with type 2 diabetes and control subjects. Salivary IL-17A was not different between subjects with and without diabetes, although levels were higher in control subjects with periodontitis. Salivary IL-18 associated with plasma glucose with serum IL-18 levels associated with HbA1c. The authors conclude that IL-17A and IL-18 have a role in periodontal inflammation and type 2 diabetes.

Please provide information on method used to collect saliva and how much was collected.

Were the saliva and serum samples run in duplicate or triplicate?

Was saliva concentrated and then the measure corrected?

Why was protease inhibitor cocktail added to saliva but not blood samples?

Controls with periodontitis have higher IL-17A than controls without, but there are only 8 subjects with periodontitis compared to 17 without. Statistical significance here is marginal. What is the likely outcome of comparing equal size groups?

Although effect size is mentioned as proof that there were enough subjects, use of the standard deviation of the control population could be used to calculate the number of subjects needed to see a significant difference in the diabetic subjects. A calculation of how many to see a 25% increase in IL-17A could be informative.

Line 486 in Conclusion does not add to the conclusion and can be removed.

6. PLOS authors have the option to publish the peer review history of their article (what does this mean?). If published, this will include your full peer review and any attached files.

Reviewer #1: No

Reviewer #2: No

---

## [Author Response · Author response to Decision Letter 0]

26 Nov 2019

Response to Reviewer #1 

 1) The introduction is long and could be reduced.

We have cut down some unnecessary information that is not important (page 2 lines 61-62 and 69-71). 

2) To be diagnosed with periodontitis how many sextants had to score 3 or 4? 

In this study, subjects with a maximum PSR score of 3 or 4 in at least one sextant were considered to have periodontitis. This sentence has been added in the subjects and methods (page 7 line 150). The PSR scores for the evaluation of periodontal disease have been used in diabetic patients in a study by Almeida and colleague [1]. PSR data indicated that periodontal disease was more frequent and more severe in diabetic patients, therefore, we used PSR scores in determining the periodontal status of the diabetic subjects in our study. 

3) There was no radiographic confirmation of bone loss?

Since we conducted the study at the Outpatient Clinic of the Endocrinology and Metabolism Section at King Chulalongkorn Memorial Hospital, the radiographic examination was not performed. However, we did measure probing depth of all teeth, six sites for each tooth and this could at least be used for periodontal examination. According to a study by Primal and colleagues in 2014 [2], the predictive potential of the PSR score for the evaluation of periodontal disease was performed. The correlation of PSR score and AAP disease categories was examined and a significant correlation between the PSR scores and periodontal disease (R2 = 0.43, p < 0.0001) was observed. PSR scored a fairly accurate predictor of AAP disease category with area under receiver-operator curve = 0.73, p < 0.0001. This reference has been added in the subjects and methods section (page 7 lines 151-157).

4) The healthy control group with periodontitis has very low numbers.

We acknowledged that there were low numbers in the healthy control group with periodontitis. We knew the flaw of our methodology regarding the sample size of the control group with periodontitis. Because participants who were healthy with no medical problems generally took good care of their oral hygiene, so we were unable to recruit a high number of participants in this group. In order to circumvent this problem, a non-parametric statistical analysis was suitably performed to compare the cytokine levels among the groups and the data were presented using medians with the 1st and 3rd quartiles. Afterwards, a stepwise multiple linear regression analysis was used to investigate the association between cytokine levels and clinical parameters without reciprocal influence of each factor and the effect of known confounding factors was taken into account. As shown in Tables 3 and 4, a stepwise multiple linear regression analysis showed that salivary IL-17A was strongly associated with periodontal disease. 

5) How and where from were blood samples collected?

Approximately 10 mL of whole blood samples were collected by venipuncture at the area of an antecubital fossa. Serum was isolated using serum-separating blood collection tubes (BD vacutainers® Plus Plastic Serum Tubes). Briefly, the collected whole blood samples were allowed to clot at room temperature for 30-45 min and the serum was isolated by centrifugation at 1000g for 15 min at 4°C. The serum samples were further aliquoted and stored at -80°C until assayed. This information has been added in the subjects and methods section (page 8 line 176-179).

6) Although a sample size calculation was undertaken it is just for two groups, not the four you have. 

We acknowledged this comment and would like to explain that at first, we would like to investigate the cytokines levels including IL-17A and IL-18 in only 2 groups of samples including the diabetic and the control groups, therefore we calculated the sample size just for 2 groups. As explained in comment number 4, as the number of subjects was low, a non-parametric statistical analysis was instead utilized. 

7) Did you also age/sex match the groups?

We did not match the age/sex of the subjects, however, the majority of subjects in all groups were female and from Chi-square tests shown in Table 1, there was no significant difference in gender among the groups.

8) I feel you need to repeat the sample size calculation and recruit more healthy periodontitis patients.

Thank you for your advice. As stated in comment number 4 we have explained the limitation of our study and have used an appropriate statistical analysis to correct this problem. 

9) You need to add exactly what the laboratory methods were.

Thank you for the suggestion. We have added how to collect saliva (page 8 lines 171-174) and serum samples (page 8 lines 176-179). The details how to investigate the levels of IL-17A and IL-18 using ELISA kits were added (page 9 lines 189-201). In addition, we have modified the statistical analysis part as recommended by the reviewer (page 10 lines 217-230). 

10) How many samples did not have detectable IL-17 or 18?

IL-17A could not be detected in a few saliva samples and this cytokine could not be detected in only 1 serum sample. These data were excluded in statistical analysis. In contrast, IL-18 could be detected in all of our saliva and serum samples.

11) The results introduce new methods with the classification by PSR. This is not what the study was designed for and may not be valid statistically.

At first, we examined the IL-17A and IL-18 levels in the saliva and serum samples and investigated whether these levels were different between the diabetic subjects and healthy controls. We found no statistical difference of the cytokine levels between the two groups. However, there were some studies which indicated that these cytokine levels might be affected by periodontal status. We then divided the diabetic and control subjects according to their periodontal conditions and found an interesting result that there was a trend of increasing IL-17A levels in the saliva samples of both control and diabetic subjects with periodontitis (Figure 2E). Furthermore, in Figure 3, we demonstrated that when grouping all subjects according to PSR scores, significant differences of salivary IL-17A levels among different groups of PSR scores were observed. We then examined the association of these cytokines and clinical parameters using stepwise multiple linear regression analyses which justified the conclusion we made. 

12) The legends for Tables 3 and 4 need to better explain what they show.

We have added that stepwise multiple linear regression analysis was used to calculate the association between clinical parameters and the levels of cytokines in Table 3 (page 16 lines 330-331, 351) and Table 4 (page 17 lines 363-364).

13) The results are long and perhaps overanalyze the data

We aimed to explain the results first by comparing the levels of cytokines between the diabetic and the control groups and then among the 4 groups of subjects (diabetic subjects with or without periodontitis and healthy controls with or without periodontitis). We also investigated the correlations between cytokine levels and clinical parameters using partial correlation analysis. Subsequently, stepwise multiple linear regression analysis was utilized. Since the relationships between various immunological parameters are complex and a cytokine may affect differently on different subjects, at different times and in the presence (or absence) of other clinical parameters, we, therefore, used multivariate statistics (in our study, stepwise multiple linear regression analysis) to examine multiple variables while controlling the effect of other variables at the same time [3].

Response to Reviewer #2 

Techatanawat et al. examined interleukin-17A and interleukin-18 in paired saliva and blood samples from subjects with type 2 diabetes and control subjects. Salivary IL-17A was not different between subjects with and without diabetes, although levels were higher in control subjects with periodontitis. Salivary IL-18 associated with plasma glucose while serum IL-18 levels associated with HbA1c. The authors conclude that IL-17A and IL-18 have a role in periodontal inflammation and type 2 diabetes.

1) Please provide information on method used to collect saliva and how much was collected.

The method used for saliva collection was added in the materials and methods section (page 8 lines 171-174). The unstimulated whole saliva was collected using the standard method described by Navazesh et al.[4]. Briefly, subjects were asked to spit the saliva approximately 5 mL into a sterile tube while placing that tube on ice. Protease inhibitor cocktail (Roche Diagnostics GmbH, Mannheim, Germany) was added immediately after saliva collection. Saliva was subsequently centrifuged at 10,000 g, 4°C for 10 min to collect only the supernatant which was further aliquoted and stored at -80°C.

2) Were the saliva and serum samples run in duplicate or triplicate?

We did not duplicate or triplicate all samples due to budget limitations, but we determined intra-assay and inter-assay coefficient of variations (CV) in our samples. Our intra-assay CV was less than 1% and inter-assay CV was 4.5%, which were quite acceptable.

3) Was saliva concentrated and then the measure corrected?

The saliva samples had not been concentrated. In fact, the concentrations of some saliva samples were too high for the measurement by ELISA and we had to dilute the saliva samples by adding the diluent provided by the manufacturer for appropriate measurement. 

4) Why was protease inhibitor cocktail added to saliva but not blood samples?

Whole saliva is a non-sterile biofluid containing substances from gingival crevicular fluid, dental plaque, and oral microbes. Thus, saliva contains proteolytic enzymes mostly derived from white blood cells and bacteria, and partly from salivary glands. Furthermore, increased saliva concentrations of proteolytic enzymes such as neutrophil collagenase MMP-8 and gelatinase MMP-9 as well as increased saliva enzyme activities of MMP-2, MMP-9 and elastase have been reported to correlate with periodontal inflammation [5-7]. These enzymes can damage other protein contents in the whole saliva including cytokines. Therefore, in saliva research especially studies in salivary proteins, it has been suggested to add protease inhibitor cocktail in saliva samples in order to preserve protein contents and minimize protein degradation in the collected saliva samples [8, 9].

On the contrary, whole blood is a sterile biofluid and serum samples were cell free and devoid of clotting factors. In our study, we collected whole blood using serum-separating blood collection tubes and separated serum samples using centrifugation. Serum samples were then immediately frozen at -80°C. These processes were completed within 2 hours after blood collection. Ayache et al. [10] demonstrated that changes in protein levels in plasma during storage did not depend on protease inhibitors but mainly from cytokines produced by leukocytes and platelets. They suggested that there is no advantage in using tubes containing protease inhibitors if we can separate plasma from whole blood and immediately freeze them. Moreover, many studies demonstrated that cytokines can be measured in serum samples without using protease inhibitors [11-13]. Therefore, in our study we did not add protease inhibitors in blood samples.

5) Controls with periodontitis have higher IL-17A than controls without, but there are only 8 subjects with periodontitis compared to 17 without. Statistical significance here is marginal. What is the likely outcome of comparing equal size groups?

We acknowledged the comment regarding the low number of subjects in the control without periodontitis group. As explained previously (reviewer number 1 comments 4 and 8), in this case we used non-parametric statistical analysis to compare the levels of cytokines among groups. From stepwise multiple linear regression analysis, after controlling the glycemic status using HbA1C and FPG, salivary IL-17A levels were still positively associated with PSR scores. Therefore, there is evidence to believe that the likely outcome of comparing equal size group would provide the same result. 

6) Although effect size is mentioned as proof that there were enough subjects, use of the standard deviation of the control population could be used to calculate the number of subjects needed to see a significant difference in the diabetic subjects. A calculation of how many to see a 25% increase in IL-17A could be informative.

Since there was no previous study regarding these cytokine levels in Thai population, we do not have the data indicating the standard deviation of the control population. We acknowledged that the subjects in some groups should be increased and if a further study is to be performed, we will try our best to recruit more subjects. 

7) Line 486 in Conclusion does not add to the conclusion and can be removed.

Thank you for pointing out this error. We have already removed the conclusion as suggested.

---

## [Decision Letter · Decision Letter 1]

28 Jan 2020

Salivary and serum interleukin-17A and interleukin-18 levels in patients with type 2 diabetes mellitus with and without periodontitis.

PONE-D-19-15328R1

Dear Dr.Siribangon Piboonniyom Khovidhunkit,,

We are pleased to inform you that your manuscript has been judged scientifically suitable for publication and will be formally accepted for publication once it complies with all outstanding technical requirements.

With kind regards,

Víctor Sánchez-Margalet

Academic Editor

PLOS ONE

Additional Editor Comments (optional):

Reviewers' comments:

Reviewer's Responses to Questions

**Comments to the Author**

1. If the authors have adequately addressed your comments raised in a previous round of review and you feel that this manuscript is now acceptable for publication, you may indicate that here to bypass the “Comments to the Author” section, enter your conflict of interest statement in the “Confidential to Editor” section, and submit your "Accept" recommendation.

Reviewer #2: All comments have been addressed

2. Is the manuscript technically sound, and do the data support the conclusions?

Reviewer #2: (No Response)

3. Has the statistical analysis been performed appropriately and rigorously? 

Reviewer #2: (No Response)

4. Have the authors made all data underlying the findings in their manuscript fully available?

Reviewer #2: (No Response)

5. Is the manuscript presented in an intelligible fashion and written in standard English?

Reviewer #2: (No Response)

6. Review Comments to the Author

Reviewer #2: (No Response)

7. PLOS authors have the option to publish the peer review history of their article (what does this mean?). If published, this will include your full peer review and any attached files.

Reviewer #2: No

---

## [Editor Report · Acceptance letter]

3 Feb 2020

PONE-D-19-15328R1 

Salivary and serum interleukin-17A and interleukin-18 levels in patients with type 2 diabetes mellitus with and without periodontitis. 

Dear Dr. Khovidhunkit:

I am pleased to inform you that your manuscript has been deemed suitable for publication in PLOS ONE. Congratulations! Your manuscript is now with our production department. 

With kind regards,

on behalf of

Dr. Víctor Sánchez-Margalet 

Academic Editor

PLOS ONE